# Study on Microstructure and Properties of Black Micro-Arc Oxidation Coating on AZ31 Magnesium Alloy by Orthogonal Experiment

**DOI:** 10.3390/ma15248755

**Published:** 2022-12-08

**Authors:** Hongzhan Li, Yifei Wang, Juanjuan Geng, Shaolong Li, Yongnan Chen

**Affiliations:** 1Northwest Institute for Nonferrous Metal Research, Xi’an 710016, China; 2Rare Mental Materials Surface Engineering Technology Research Center of Shaanxi Province, Xi’an 710016, China; 3School of Material Science and Engineering, Northeastern University, Shenyang 110819, China; 4School of Materials Science and Engineering, Chang’an University, Xi’an 710064, China

**Keywords:** AZ31B magnesium alloy, micro-arc oxidation, black MAO coating, orthogonal experiment, corrosion resistance, hemisphere emissivity

## Abstract

The effects of CuSO_4_ concentration, voltage and treating time on the hemisphere emissivity and corrosion resistance of AZ31B magnesium-alloy black micro-arc oxidation coatings were studied by orthogonal experiment. The microstructure, phase composition, corrosion resistance and hemisphere emissivity of the coating were investigated by scanning electron microscopy, energy-dispersive X-ray spectroscopy, X-ray diffraction, X-ray photoelectron spectroscopy, electrochemical test and infrared emissivity spectrometer, respectively. The results showed that the influences of each factor on corrosion current density and the hemisphere emissivity are as follows: voltage > treating time > CuSO_4_ concentration. The black MAO coatings are mainly composed of WO_3_, MgAl_2_O_4_, CuAl_2_O_4_, MgO, CuO and MgF_2_. The CuO and CuAl_2_O_4_ phases are the main reasons for blackness of the coatings. The coating exhibits the best corrosion resistance under the conditions of CuSO_4_ concentration 1.5 g/L, oxidation voltage 500 V and treating time 10 min. Additionally, the variation trends of hemispherical emissivity and roughness of the black MAO coating are the same when the composition of the coatings is similar. When the concentration of CuSO_4_ is 1.5 g/L, the oxidation voltage is 450 V and the treatment time is 10 min, the coating with the highest hemispherical emissivity of 0.84 can be obtained.

## 1. Introduction

Magnesium alloys are widely used in aerospace, transportation, 3C electronic products, biomedical, energy sectors, railroad industries, etc., due to their low density, high strength, excellent machining and shock absorption performance, as well as good electromagnetic shielding and damping performance. They are considered as the green engineering material of the 21st century [1,2,3]. However, the poor chemical stability, low electrode potential and inferior corrosion resistance of magnesium alloys have greatly limited their applications. Due to these limitations, it is necessary to carry out proper surface treatment on magnesium alloys before they are put into industrial applications.

Micro-arc Oxidation (MAO) is an effective and eco-friendly surface modification technique which can provide ceramic coatings on non-ferrous metal (Mg, Al, Ti, Zr and its alloys) surfaces with high wear resistance, corrosion resistance and great hardness [4,5,6]. The MAO process generates a micro-arc discharge on the surface of the material through a high-voltage electrical breakdown, which mixes plasma chemistry and electrochemistry oxidation with a spark treatment that has a high voltage in an alkaline electrolyte, leading to the formation of ceramic coatings on the metal surface [7,8,9,10].

In recent years, with the development of technology in the aerospace field, the preparation of black MAO coating with excellent thermal control properties on the surface of magnesium alloys by micro-arc oxidation technology has gradually attracted the attention of researchers [11,12,13]. It is evident that many black MAO coatings with high thermal control properties have been prepared on Ti, Al, Mg and their alloys. Yao et al. [14] investigated the black thermal control ceramic coating on the surface of the Ti6Al4V alloy by plasma electrolytic oxidation. It was found that the high absorption (>0.95) and emission (>0.92) were attained by PEO with the coloring additives of NH_4_VO_3_, FeSO_4_ and Co(CH_3_COO)_2_. Ramakrishnan et al. [15] studied the effect of nanoparticle additives (CuO, TiO_2_) in an aqueous electrolyte on the thermal control property of PEO coating on AA7075. It was reported that the CuO particles incorporated in PEO coating showed the highest thermal diffusivity (80.18 mm^2^/s at 30 °C) compared to other PEO-coated samples. He et al. [16] reported that the black MAO films formed on the AZ91 alloy may mainly be owing to the existence of CuO particulates in the porous outer layer after the addition of Cu_2_P_2_O_7_ to silicate electrolytes. The research of Lu et al. [17] indicates that the solar absorptance and infrared emittance of MAO coating increased initially with the increase in the concentration of Fe_2_(SO_4_)_3_. It was claimed that at the Fe_2_(SO_4_)_3_ concentration of 8 g L^−1^, the coating has its highest solar absorptance (0.94) and infrared emissivity (0.83). However, most of the studies at present focus on the effect of electrolyte additives or electrical parameters on the absorptance and emissivity of MAO coating [18,19,20,21,22]. Moreover, the effects of additives and electrical parameters on the microstructure, morphology and phase composition of magnesium-alloy black MAO coating on thermal control performance still need to be systematically investigated.

In the present work, CuSO_4_ was selected as an additive to prepare black ceramic coatings with high thermal control performance on AZ31 magnesium alloy by MAO. The effects of CuSO_4_ concentration, voltage and treating time on surface structure, element content, chemical composition, emissivity and corrosion resistance of black MAO coating were investigated through orthogonal experiment.

## 2. Materials and Methods

### 2.1. Samples and Coating Preparation Procedures

The AZ31B magnesium alloy (Al 2.86 wt.%, Zn 0.84 wt.%, Mn 0.29 wt.%, Si 0.01 wt.%, Cu < 0.01 wt.%, Ni < 0.001 wt.%, Fe < 0.003 wt.%, Ca < 0.01 wt.% and Mg balance) was used as the substrate material in this study and cut in a size of 15 × 15 × 5 mm to fabricate MAO coatings. Before MAO treatment, the surface of the samples was ground and polished with SiC abrasive paper progressively from 800 to 2000 grit, subsequently cleaned ultrasonically with ethanol and deionized water, and then dried.

### 2.2. Experiment Process

The MAO process was carried out in T-25Z MAO equipment (Northwest Institute for Nonferrous Metal Research, Xi’an, China) with a frequency of 500 Hz and a duty cycle of 15%, in a constant voltage mode. The AZ31B magnesium alloy substrate and the stainless-steel sheet were used as anodes and cathodes in the MAO process. Potassium sodium tartrate (C_4_H_4_KNaO_6_), NaF, CuSO_4_, NaAlO_2_ and Na_2_WO_3_ were the main components of the basic electrolyte, and NaOH was mainly used to adjust the pH value of the solution (10–11). CuSO_4_ was chosen as one of the orthogonal factors. Using L_9_(3^4^) standard orthogonal form, factors and levels’ design are shown in Table 1. During the MAO process, the temperature of the electrolyte is maintained below 25 °C through a stirring and cooling system. After the MAO treatment, the MAO coatings were washed using deionized water and dried with cool air.

### 2.3. Characterization

The surface and cross-sectional morphologies of the coatings were examined by scanning electron microscopy VEGA II XMU (OXFORD, Oxford, UK), and the element distribution on the surface and cross-section were analyzed by energy dispersive spectrum (EDS). The three-dimensional morphology and roughness of the coatings were examined by laser scanning confocal microscopy (Zeiss LSM 800; Zeiss, Germany). The phase composition of coatings was performed by D/max 2200 PC X-ray diffractometer (RIGAKU, Tokyo, Japan) with Cu K radiation at 45 kV and 200 mA, and the detected range was between 20 and 80 with a step size of 0.02 and a scanning rate of 2/min. The chemical composition of the coatings was determined by X-ray photoelectron spectroscopy (XPS, ESCALAB 250Xi; Thermo Fisher Scientific, Waltham, MA, USA). Surface porosity of the MAO coatings was measured by Image J software. The hemisphere emissivity in the range of 3 μm–30 μm of the coatings was measured by an AE1 portable infrared emissivity spectrometer (D&S, Norwalk, CT, USA) at room temperature.

An electrochemical test was conducted in a 3.5 wt.% NaCl solution at room temperature with electrochemical workstation Versa STAT 3F (AMETEK, Pennsylvania, PA, USA). During electrochemical tests, a saturated calomel electrode (SCE) was used as a reference electrode, the MAO coated sample as a working electrode (1 cm^2^ exposed area), and a Pt grid as a counter electrode. The polarization curves were measured at a scanning rate of 2 mV/s, with the potential range of −0.5 V to 2.0 V. The corrosion parameters such as anodic slopes (b_a_) and catholic slopes (−b_c_) were extracted by the Tafel extrapolation method. Polarization resistance (R_p_) was calculated using the Stern–Geary Equation (1) which represents the corrosion property of black MAO coating.
(1)$$R_{p}=\frac{b_{a}\cdot \left|b_{c}\right|}{2.303i_{\mathrm{corr}}\left(b_{a}+\left|b_{c}\right|\right)}$$

## 3. Results

### 3.1. Orthogonal Experimental Results and Analysis

Table 2 shows the results of the orthogonal experiment collected via the range analysis approach used to explore each factors’ effect and its interaction on the responses. Namely, corrosion current density (I*_corr_*), the hemisphere emissivity and porosity were used as measures of coating quality. In Table 2, k1, k2 and k3 represent the mean values of levels 1, 2 and 3 of the three factors, respectively. The ‘R’ is the difference between maximum and minimum values among k1, k2 and k3. The greater the R is, the more important the factor is. Therefore, based on R values, the influences of each factor and their interactions on the corrosion current density (I*_corr_*), the hemisphere emissivity and the porosity of the coatings are in the same order and could be ranked as follows: voltage > treating time > CuSO_4_ concentration, and the best scheme is A2B3C2, A2B3C3 and A2B1C3, respectively.

In conclusion, the optimal process schemes corresponding to different evaluation indicators are different. However, considering the main application goals of magnesium-alloy black MAO coating, that is, the hemisphere emissivity and corrosion resistance, I*_corr_* and the hemisphere emissivity should be the primary factors considered. Under the same electrolyte concentration, I*_corr_* decreases and the hemisphere emissivity increases with the increase in oxidation voltage, and the change in porosity of the coating is not obvious. Considering the application requirements of the coating, the best process scheme for the preparation of black MAO coating is A2B3C2.

### 3.2. Surface Morphologies and Elemental Composition of Black MAO Coating

Figure 1 shows the digital photographs and SEM images of the black MAO coating formed based on the orthogonal experiment. It can be seen from Figure 1 that with the increase in oxidation voltage and time, the coating of samples 1^#^–3^#^ has a similar volcanic pore structure, and the pore size increases significantly. Despite the change of process conditions, the coatings of samples 4^#^–9^#^ have similar microscopic morphology with a typical uneven micropore structure, some protuberances and a small number of microcracks. It is obvious that all the coatings of the sample present the typical surface morphologies of the MAO coatings [23]. According to the digital photographs at the upper right corner of each sample in Figure 1, the blackness of the coatings is relatively similar. The results show that the increase in the CuSO_4_ concentration in the electrolyte, reaction time and reaction voltage has no obvious effect on the blackness of the coating.

The SEM surface micrographs and corresponding elemental distribution of sample 5^#^ are shown in Figure 2. Table 3 shows the EDS results of the coating surface (1^#^–9^#^) prepared in different parameters. It can be seen from Figure 2 and Table 3 that the coating contains Mg, Al, O, Cu, F and W elements, in which Mg and part of Al are from the substrate, and O, Cu, F and W originate from the electrolyte, indicating that elements from the substrate and electrolyte all participate in the coating formation reaction. Obviously, most elements of coating come from the electrolyte. When CuSO_4_ is doped into the electrolyte, the element Cu comes into the coating and forms a copper oxide, thus promoting the formation of black MAO coating. This finding is in accordance with the result reported by Wu et al. [24]. 

Figure 3 shows the 3D surface morphologies and depth profiles of black MAO coating samples 1^#^–9^#^. The surface roughness (Ra) of black MAO coating samples 1^#^–9^#^ are given in Figure 4. Confocal microscope images (Figure 3) exhibit the change of black MAO coating surface morphology under different process conditions. The 3D surface morphologies of the coating show lots of pores, grooves and protrusions. The average surface roughness parameter values (Ra) of samples 1^#^−9^#^ are 1.09 μm, 2.12 μm, 2.73 μm, 0.75 μm, 2.41 μm, 2.13 μm, 0.72 μm, 1.95 μm, 2.89 μm, respectively. As shown in Figure 3, when the oxidation voltage increases from 400 V to 500 V, the coating surface becomes rougher. This is because the arc discharge generated by the increase in oxidation voltage during micro-arc oxidation is more intense, which increases the number of coating pores and protrusions. The change trends of the surface morphology and roughness of the coating in Figure 3 (4^#^−6^#^) and Figure 3 (7^#^−9^#^) is similar to that in Figure 3 (1^#^−3^#^). That is, at the same CuSO_4_ concentration, the surface roughness of the coating increases significantly with the increase in the voltage. Correspondingly, the porosity of the coating in Table 2 also indicates similar changes. The increase in surface roughness of MAO coatings depends on the formation of coatings with larger pores, microcracks, grooves and protrusions [25,26].

### 3.3. Cross-Sectional Morphologies and Elemental Composition of Black MAO Coating

The cross-sectional morphologies of black MAO coating samples 1^#^−9^#^ are shown in Figure 5. Overall, the coatings consist of an inner layer and an outer layer. The inner layer of the coating is dense, almost without micropores while the outer layer of the coating has obvious micropores and microcracks. This indicates that the outer layer of the coating is loose. From samples 1^#^, 2^#^ and 3^#^, it can be found that the coating becomes thicker with the increase in voltage, and the micropores and cracks in the outer layer of the coating become larger with the increase in voltage and time. Comparing the samples 2^#^, 5^#^ and 8^#^, it can be deduced that by increasing the CuSO_4_ concentration to 450 V, the thickness of the coating and the number of micropores in the outer layer of the coating increase significantly. The comparison of samples 1^#^–9^#^ indicates that compared to the content of CuSO_4_, as well as the voltage and time on the thickness of the coating, the formation of micropores and cracks has a greater impact. This is consistent with the conclusion that electric parameters affect the thickness, micropores and cracks of MAO coatings in other electrolyte systems [27,28].

The SEM cross-sectional micrographs and corresponding elemental distribution of sample 5^#^ are shown in Figure 6. It is clearly observed that in the coating, the main elements are Al, Mg, O, Cu, F and W. The presence of Mg originates from the substrate, whereas O, Cu, F, W and most of Al are originated from the electrolyte. As can be seen, each element possessed a relatively uniform distribution across the sample section.

### 3.4. Phase and Chemical Composition of Black MAO Coating

The X-ray diffraction patterns for black MAO coating samples 1^#^–9^#^ are shown in Figure 7. As shown in the figure, the black MAO coatings are predominantly composed of WO_3_, MgAl_2_O_4_, CuAl_2_O_4_, MgO and MgF_2_ phases. In the angle range of 10°–30°, each coating sample has the characteristics of low crystallinity and a high diffraction peak background, and its phases are mainly WO_3_, MgAl_2_O_4_ and CuAl_2_O_4_. No diffraction peaks corresponded to CuO, making the coating black [29]. Therefore, more characterization methods need to be used to analyze the composition of coatings.

Figure 8 shows the high-resolution spectra of major elements in the black MAO coating of samples 1^#^–9^#^. The Al 2p spectrum in Figure 8a can be divided into two peaks, 74.1 eV and 74.7 eV, which are attributed to MgAl_2_O_4_ and CuAl_2_O_4_, respectively. Figure 8b shows that Cu 2p binding energies at 933.2 eV and 935.0 eV are the typical binding energies for CuO and CuAl_2_O_4_ [30,31]. As shown in Figure 8c, the Mg 1s spectrum shows three peaks at 1303.9, 1304 and 1306.4 eV, corresponding to MgO, MgAl_2_O_4_ and MgF_2_ [32], respectively. In Figure 8d, the spectrum of O 1s exhibited multiple peaks, at 530.8, 531.4, 531.5 and 532.1 eV, corresponding to WO_3_, CuAl_2_O_4_, MgAl_2_O_4_ and MgO [33,34], respectively. Figure 8e shows that W 4f binding energy at 35.4 and 37.9 eV corresponds to WO_3_ [35]. Compared with the results of XRD, the presence of CuO in the coating was confirmed by XPS.

### 3.5. Electrochemical Corrosion Test of Black MAO Coating

Figure 9 exhibits the potentiodynamic polarization curves of AZ31B magnesium alloy and black MAO coating samples 1^#^–9^#^ in 3.5 wt.% NaCl solution. The fitting results are shown in Table 4. It can be seen from Figure 9 and Table 4 that the E*_corr_* and I*_corr_* of AZ31B substrate were determined to be −1.522 V and 2.582 × 10^−4^ A/cm^2^, respectively. After the MAO treatment, the black MAO coating samples 1^#^–9^#^ showed a positive increment in E*_corr_* and a negative increment in I*_corr_* compared to those of the AZ31B substrate. The lowest value of I*_corr_* (1.536 × 10^−6^ A/cm^2^) and the most positive value of E*_corr_* (−1.371 V) were found in the case of sample 6^#^. The R_p_ value of sample 6^#^ is 1.915 × 10^4^, which is three orders of magnitude higher than that of the AZ31B substrate. Thus, the corrosion resistance of sample 6^#^ was superior to other cases. The corrosion resistance of the black MAO coating is slightly different under different process conditions, but it can still provide excellent protection for the AZ31B substrate.

### 3.6. Hemisphere Emissivity of the Black MAO Coating

Figure 10 shows the hemisphere emissivity of black MAO coating samples 1^#^–9^#^. As can be seen in this figure, the experiment results showed that the hemispherical emissivity value of samples 3^#^, 5^#^ and 9^#^ are 0.83, 0.84 and 0.83, respectively, which are greater than 0.8. When the content of CuSO_4_ is 1 g/L, the sample’s hemispherical emissivity increases with the oxidation voltage and time. However, when the content of CuSO_4_ increases to 1.5 g/L and 2 g/L, respectively, the hemisphere emissivity of samples under different process parameters does not increase significantly. According to the infrared radiation theory, the hemispherical emissivity of the coating is mainly affected by its thickness, surface structure and composition [36]. Since the coating composition of samples 1^#^–9^#^ under experimental conditions is almost the same, the hemispherical emissivity may be determined by the coating structure (thickness and roughness). In comparing Figure 4 and Figure 10, it can be found that the surface roughness (Ra) of black MAO coating samples 1^#^–9^#^ is basically the same as the change of the hemisphere emissivity. Similarly, the law of changes in the thickness of the black MAO coating samples 1^#^–9^#^ in Figure 5 is basically consistent with the change of the hemisphere emissivity.

## 4. Discussion

The MAO process, including a series of electrochemical, plasma chemical and thermal chemical reactions, is an essential process. In this work, the black MAO coating on the AZ31B alloy has typical porous structures and the main phase compositions are WO_3_, MgAl_2_O_4_, CuAl_2_O_4_, MgO, CuO and MgF_2_. The formation process of the main phase in the black MAO coating, as shown in Equations (2)–(8) [28,37] are as follows:Mg → Mg^2+^ + 2e(2)
Mg^2+^ + 2OH^−^ → Mg(OH)_2_(3)
Mg(OH)_2_ → MgO + H_2_O(4)
Mg^2+^ + 2F^−^ → MgF_2_(5)
2WO_4_^2−^ − 4e → 2WO_3_ + O_2_↑(6)
Cu^2+^ + 2OH^−^ → Cu(OH)_2_(7)
Cu(OH)_2_ → CuO + H_2_O(8)

Meanwhile, aluminate ions are hydrolyzed to form the aluminum hydroxide (Equation (9)), and aluminum hydroxide is decomposed to form Al_2_O_3_ (Equation (10)). The partially molten Al_2_O_3_ reacts with MgO and CuO to produce MgAl_2_O_4_ (Equation (11)) and CuAl_2_O_4_ (Equation (12)), respectively [38,39]:AlO^2−^ + 2H_2_O → Al(OH)_3_ +OH^−^(9)
2Al(OH)_3_
→ Al_2_O_3_ + 3H_2_O(10)
MgO + Al_2_O_3_
→ MgAl_2_O_4_(11)
CuO + Al_2_O_3_
→ CuAl_2_O_4_(12)

Combined with the previous analysis, it can be speculated that the formation of CuO and CuAl_2_O_4_ in the coating is the main reason for the coating to show black. This shows that the colorant (CuSO_4_) is crucial to the black of the MAO coating on the AZ31B magnesium alloy. The digital photographs in Figure 1 indicate that the oxidation voltage and oxidation time have less impact on the color difference of the black MAO coating samples 1^#^–9^#^.

The corrosion resistance of the coating is affected by many factors, such as microstructure, thickness, roughness porosity, etc. Due to different oxidation voltage and time conditions, the thickness and structure of the black MAO coating formed on the magnesium alloy substrate are different. Therefore, the corrosion resistance of each black MAO coating sample shows its own difference. Among all coating samples, the black MAO coating sample 6^#^ shows the best corrosion resistance due to its relatively fewer structural defects and low porosity.

It can be seen from Table 2 and Figure 4 that the black MAO coating has the best roughness when the oxidation voltage is low. However, high oxidation voltage can lead to an increase in the intensity of arc discharge, which will increase the driving force during the MAO reaction process, thereby appearing in large-sized micropores and cracks in the coating, resulting in increased coating roughness. Since the hemispherical emissivity of the coating is related to the roughness, the hemispherical emissivity can be improved by adjusting the roughness of the coating when the coating composition is determined. However, excessive roughness will lead to the decreased corrosion resistance of the coating. Therefore, for engineering applications, the balance between corrosion resistance and hemispherical emissivity of the coating needs to be further considered.

## 5. Conclusions

In this work, a black micro-arc oxidation coating on the surface of AZ31B magnesium alloy was prepared by orthogonal experiment. The microstructure, phase composition, corrosion behavior and hemispherical emissivity of the black MAO coating were studied. The main conclusions are as follows:The results of the orthogonal experiment showed that the influences of each factor on corrosion current density and the hemisphere emissivity are as follows: voltage > treating time > CuSO_4_ concentration. The CuSO_4_ concentration, voltage and reaction time have no obvious effect on the blackness of the coating. The best process scheme for the preparation of black MAO coating is A2B3C2, which means the parameter combination is 1.5 g/L CuSO_4_, oxidation voltage is 500 V and treating time is 10 min;The black MAO coatings on the AZ31B magnesium alloy have a typical porous structure and black appearance. The coatings are mainly composed of WO_3_, MgAl_2_O_4_, CuAl_2_O_4_, MgO, CuO and MgF_2_. The black color of the coating is mainly caused by CuO and CuAl_2_O_4_ phases;Sample 6^#^ has the best corrosion resistance. The corrosion current density of sample 6^#^ is two magnitudes lower than the substrate, and the polarization resistance value is three orders of magnitude higher than the substrate;The variation trends of hemispherical emissivity and roughness of the black MAO coating are the same when the composition of the coatings is similar. The process conditions are 1.5 g/L CuSO_4_, 450 V of oxidation voltage and 20 min of treating time, and the obtained coating possessed the highest hemispherical emissivity of 0.84.

In general, the experimental results show that increasing the surface roughness of the magnesium-alloy black MAO coating can improve its hemispherical emissivity. This discovery will be conducive to the application of magnesium-alloy black MAO coating in spacecraft-radiation heat dissipation. However, the stability under the long-term sunlight of the hemisphere emissivity of the magnesium-alloy black MAO coating needs to be further studied. At the same time, in-depth research on improving the density of magnesium-alloy black MAO coating to improve its abrasion resistance and corrosion resistance without reducing the hemisphere emissivity is worthwhile.

## Figures and Tables

**Figure 1 materials-15-08755-f001:**
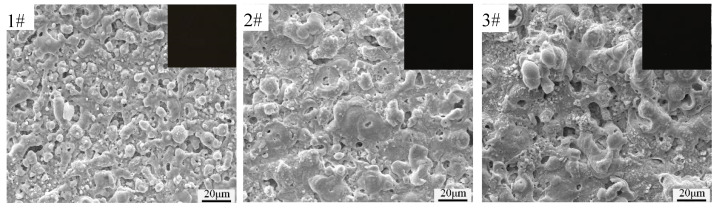
SEM surface micrographs and digital photographs (in the upper right corner of each photo) of black MAO coating samples 1^#^−9^#^.

**Figure 2 materials-15-08755-f002:**
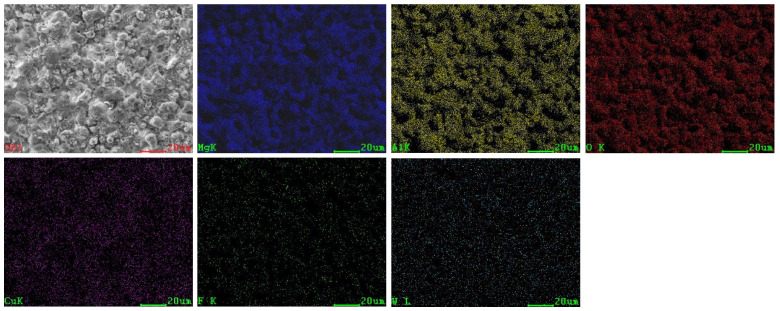
SEM surface micrographs and corresponding elemental distribution of sample 5^#^.

**Figure 3 materials-15-08755-f003:**
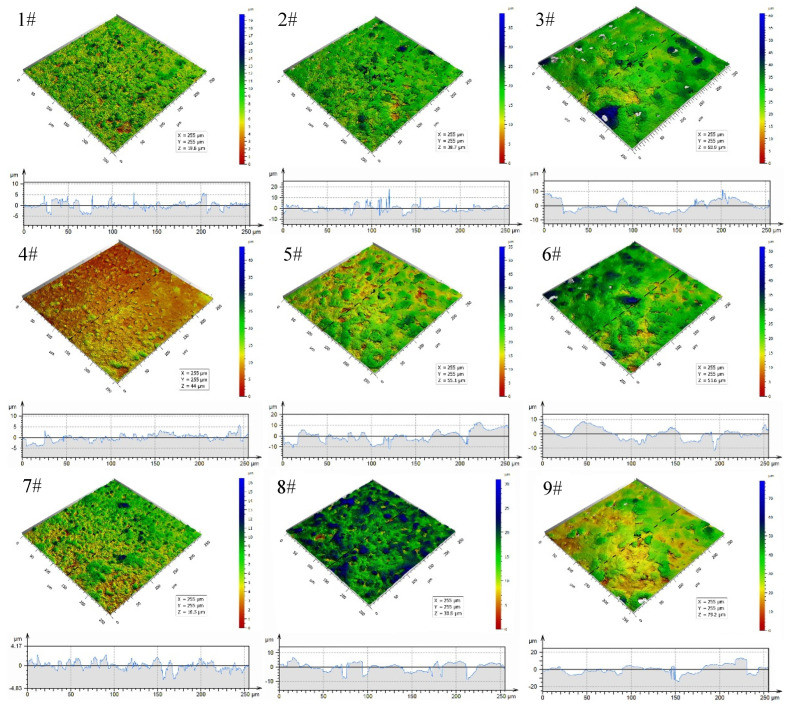
3D surface morphologies and depth profiles of black MAO coating samples 1^#^−9^#^.

**Figure 4 materials-15-08755-f004:**
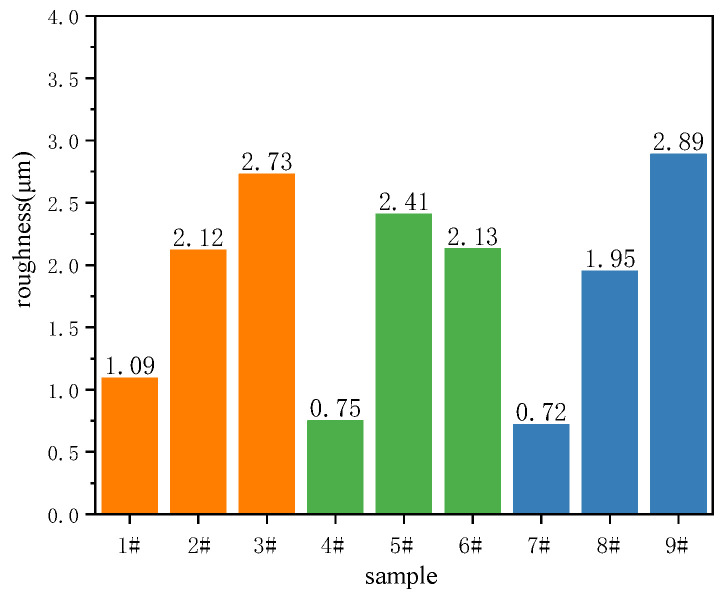
Surface roughness (Ra) of black MAO coating samples 1^#^−9^#^.

**Figure 5 materials-15-08755-f005:**
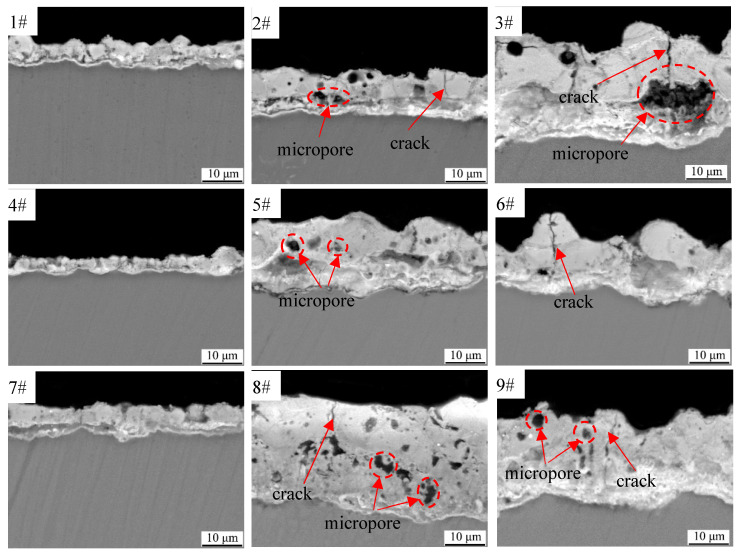
Cross-sectional morphologies of black MAO coating samples 1^#^−9^#^.

**Figure 6 materials-15-08755-f006:**
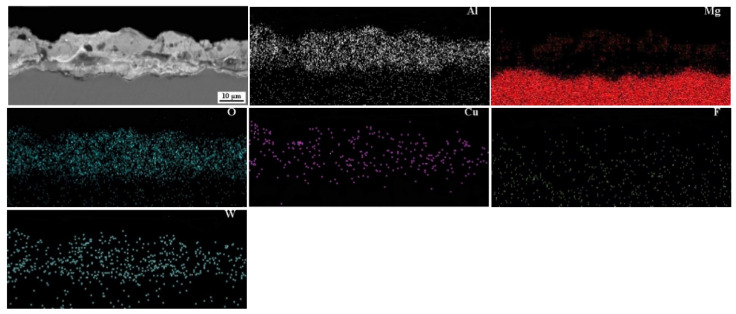
SEM cross-sectional micrographs and corresponding elemental distribution of sample 5^#^.

**Figure 7 materials-15-08755-f007:**
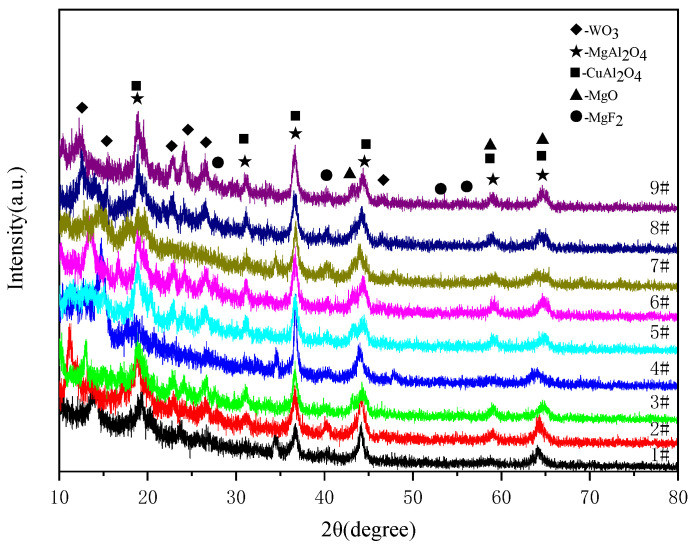
The XRD patterns of black MAO coating samples 1^#^–9^#^.

**Figure 8 materials-15-08755-f008:**
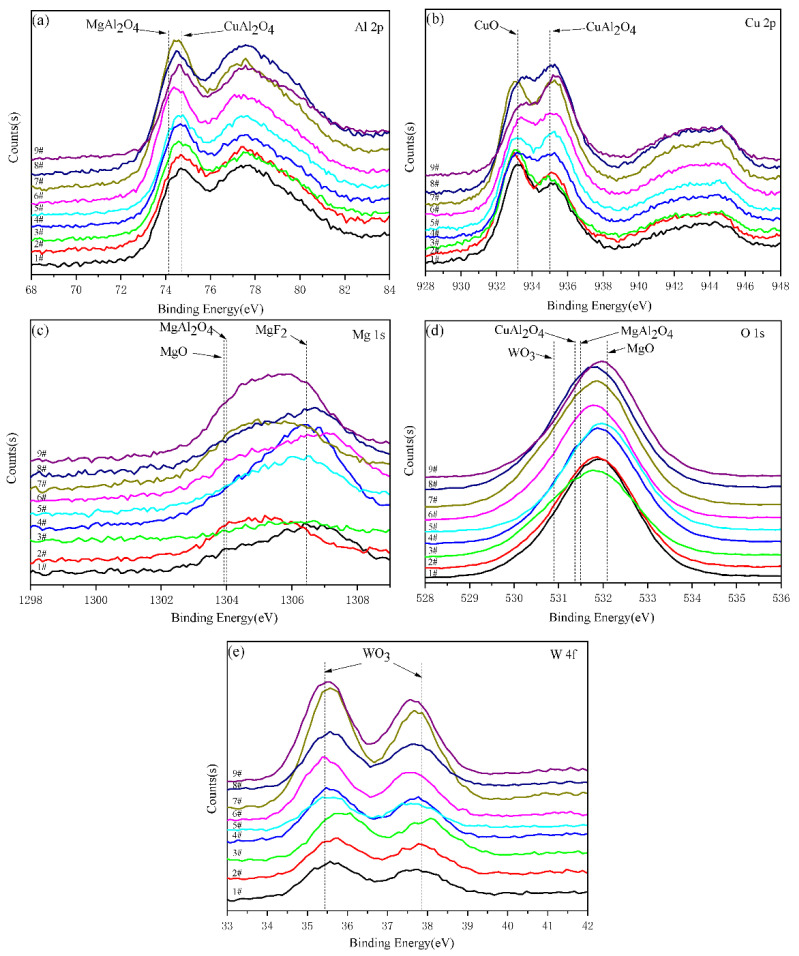
XPS high-resolution spectra of Al 2p, Cu 2p, Mg 1s, O 1s and W 4f of black MAO coating samples 1^#^–9^#^, (**a**) Al 2p; (**b**) Cu 2p; (**c**) Mg 1s; (**d**) O 1s; (**e**) W 4f.

**Figure 9 materials-15-08755-f009:**
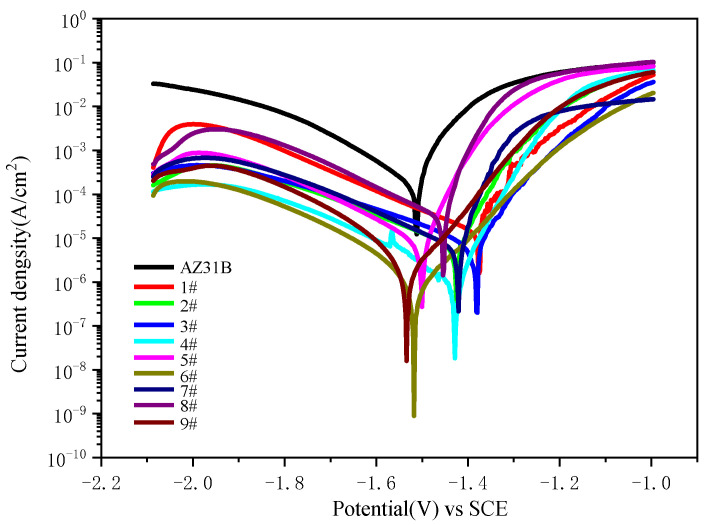
Potentiodynamic polarization curves of AZ31B magnesium alloy and black MAO coating samples 1^#^–9^#^.

**Figure 10 materials-15-08755-f010:**
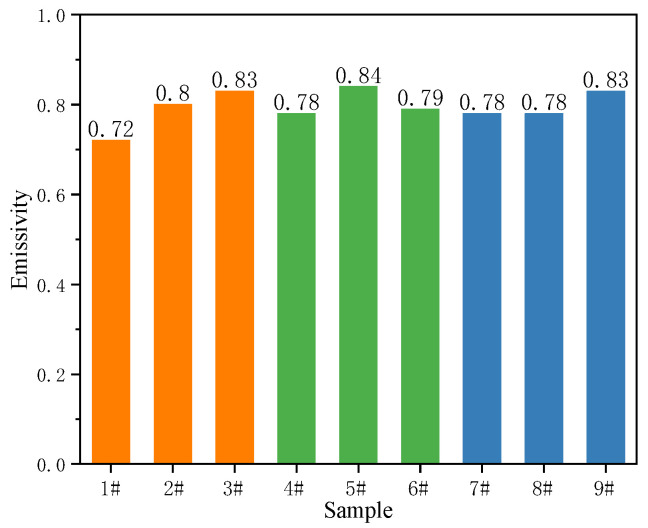
The hemisphere emissivity of black MAO coating samples 1^#^–9^#^.

**Table 1 materials-15-08755-t001:** Factors and levels of the orthogonal experiment.

Factors	A	B	C
Levels	CuSO_4_ Concentration (g/L)	Voltage (V)	Treating Time (min)
1	1	400	10
2	1.5	450	15
3	2	500	20

**Table 2 materials-15-08755-t002:** Orthogonal experimental array and analysis of experimental results.

Experimental	A	B	C	I*_corr_* (mA/cm^2^)	The Hemisphere Emissivity ε_H_	Porosity (%)
1^#^	1	400	10	0.019	0.72	6.27
2^#^	1	450	15	0.009	0.80	6.90
3^#^	1	500	20	0.011	0.83	7.19
4^#^	1.5	400	15	0.006	0.78	5.96
5^#^	1.5	450	20	0.029	0.84	6.79
6^#^	1.5	500	10	0.001	0.79	6.59
7^#^	2	400	20	0.006	0.78	5.28
8^#^	2	450	10	0.035	0.78	7.24
9^#^	2	500	15	0.002	0.83	7.58
k1	0.013 (0.783) (6.787)	0.010 (0.760) (5.837)	0.018 (0.763) (6.700)			
k2	0.012 (0.803) (6.447)	0.024 (0.807) (6.977)	0.006 (0.803) (6.813)			
k3	0.014 (0.797) (6.700)	0.005 (0.817) (7.120)	0.015 (0.817) (6.420)			
R	0.002 (0.020) (0.340)	0.019 (0.057) (1.283)	0.012 (0.054) (0.393)			

**Table 3 materials-15-08755-t003:** EDS results of the coating surface (1^#^–9^#^) prepared in different parameters.

Coating	Mg/at%	Al/at%	O/at%	Cu/at%	F/at%	W/at%
1^#^	14.00	30.43	46.99	4.16	2.84	1.58
2^#^	16.98	28.51	45.17	4.43	2.78	2.11
3^#^	21.65	24.55	44.72	4.04	2.44	2.60
4^#^	27.39	25.21	39.98	3.40	2.21	1.81
5^#^	24.23	24.34	40.44	6.88	1.03	3.09
6^#^	22.52	26.44	42.64	3.89	1.66	2.85
7^#^	25.81	22.65	44.23	2.49	2.35	2.47
8^#^	19.97	25.38	45.67	4.36	1.60	3.01
9^#^	20.52	25.96	42.73	6.27	1.78	2.74

**Table 4 materials-15-08755-t004:** Parameter values of potentiodynamic polarization curves of AZ31B magnesium alloy and black MAO coating samples 1^#^–9^#^ in 3.5 wt.% NaCl solution.

Coating	E*_corr_* (V)	I*_corr_* (A/cm^2^)	Β_a_ (mV/dec)	−β_c_ (mV/dec)	R_p_ (kΩ·cm^2^)
1^#^	−1.377	1.910 × 10^−5^	99.08	230.75	1.528 × 10^3^
2^#^	−1.377	8.973 × 10^−6^	73.07	276.68	2.798 × 10^3^
3^#^	−1.362	1.052 × 10^−5^	99.53	314.82	3.118 × 10^3^
4^#^	−1.387	6.109 × 10^−6^	53.93	158.72	2.861 × 10^3^
5^#^	−1.487	2.894 × 10^−5^	57.46	208.19	6.754 × 10^2^
6^#^	−1.371	1.536 × 10^−6^	101.43	203.87	1.915 × 10^4^
7^#^	−1.419	6.394 × 10^−6^	44.63	219.91	2.519 × 10^3^
8^#^	−1.422	3.492 × 10^−5^	43.36	218.22	4.497 × 10^3^
9^#^	−1.424	2.595 × 10^−6^	88.08	144.4	9.151 × 10^3^
AZ31	−1.522	2.582 × 10^−4^	72.22	187.41	8.767 × 10^1^

## Data Availability

Data sharing not applicable.

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
