# Peer review of "Study on Microstructure and Properties of Black Micro-Arc Oxidation Coating on AZ31 Magnesium Alloy by Orthogonal Experiment"

_materials, 2022, doi:10.3390/ma15248755_

Round 1
Reviewer 1 Report
The paper concerns the development of the design of an orthogonal experiment and the subsequent synthesis of oxide on the surface of the samples and further investigation of the syntesis products. The tasks of the article are relevant, the methods used are modern. The manuscript undoubtedly has the potential for publication in the Materials, but with mandatory revision taking into account all major comments.
We have to note that the manuscript was written rather carelessly, from where many inaccuracies appeared, it is recommended that the authors carefully double-check the entire text, drawings, samples. For example, Lines 127, 137.
The current manuscript presentation is disorganized and confusing. It is not clear from the Title: the Orthogonal Experiment refers to obtaining or study ceramic films. The authors, perhaps, do not understand the difference between the Micro-arc Oxidation process and the design of experiment. There are no arguments for choosing design factors and levels. Why the authors do not consider the influence of frequency, aluminate additives, substrate content, the choice of initial conditions and the interval is not justified?
The experimental technique is poorly described. There is no illustration of Micro-Arc Oxidation equipment system. Concentrations (or quantities of components), pH are not specified. The formula of sodium tungstate is not correct.
· Line 83-86. The MAO process was carried out in T-25Z MAO equipment (Northwest Institute for Nonferrous Metal Research, Xi’an, China) with a frequency of 500 Hz, a duty cycle of 15%.In the case of MAO process, the AZ31B magnesium alloy substrate as well as the stainless steel sheet were connected as the anode and cathode, respectively.
· Line 86-88. Potassium sodium tartrate(C4H4KNaO6), NaF, CuSO4, NaAlO2, NaWO3 were the main component of basic electrolyte, and CuSO4 was chosen as one of the orthogonal factors.
Line 280-287. The articles [38, 29] considered the effect of the quantity of sodium hydroxide on the properties of the coating. In the reviewed manuscript, sodium hydroxide is not included in the electrolyte, so equations 1-7 do not correspond to the actual processes.
It is not clear what the underlining on the Lines 99 and 100 means.
Line 57: check the correctness of the time designation and an extra space after 30.
The designation of the samples should be brought in accordance with Table 1 (Line 127, 136, 325)
Fig.1. The dimension scale is poorly visible. The caption to the figure should contain more explanations. Probably, since all samples are listed in order, there is no need to enter letter designations, it is better for the reader to indicate the sample number in the picture field instead of letter designations. What is this black square present in the upper right corner of each photos?
Fig.2. The dimension scale is poorly visible. The name of the element is almost invisible (what is the difference between the bottom three pictures?). The caption to the picture should contain more explanations.
Fig. 3. Low resolution is used. The scale on the axes is not visible. Probably, since all samples are listed in order, there is no need to enter letter designations, it is better for the reader to indicate the sample number in the picture field instead of letter designations.
Fig.5. It is not clear why sample 6 is better than 8 and 9? It also has cracks and micropores (not specified by the authors). Visually, sample 8 has a thicker oxide layer, and sample 9 oxide is denser. The same as in other photos: there is probably no need to enter letter designations, it is better for the reader to indicate the sample number in the field of the drawing instead of letter designations. The dimension scale is poorly visible.
Fig.6. It is not entirely clear why this analysis is given only for sample 5. It is more logical (based on the conclusions) to give this for the best sample (sample 6).
Fig.7. Corrosion curves look clear and not interfere with each other. It is easy to determine from them what the authors mean in the text of the article. But the color of the lines for samples 7,8,9 (in the notation in the lower left corner) is difficult to distinguish, these lines (for all samples) must be made thicker (as in the corresponding curves). The Current density signature contains an error. And for "2" it should be in uppercase.
Fig.7 and 8: Check the signature on the ordinate-axis.
The Сonclusions are rather local and relate rather only to experimental results. They are poorly correlated with the tasks specified in the Introduction and do not correspond to the title of the paper. In particular, there is not a word about the orthogonal experiment in the Conclusions.
Taking into account all the comments, the Abstract should be revised.
The country that issued the grant is not specified.
Reviewer 2 Report
Your conclusions are fine within the parameters of your study. However, you could perhaps have expanded upon that range in the Discussion by suggesting which parameters could be evaluated at a larger range in the future to increase the efficacy of these coatings. The correlations that you found were fine but none of them were particularly strong, pointing toward major advances in this technology. You should perhaps add a short paragraph on where this research goes next.
You mention wear resistance in the Introduction but then there is no other mention. Looking at the macro/micro structure of these coatings, I would guess that they would have sub-standard wear resistance. Is there potential for research on increasing density to increase wear resistance? I am just trying to put out some ideas for and end paragraph to the discussion section that deals with where you go next, because while this work is fine, it is not enough to show that these coatings in their current form have a lot of applications. More work needs to be done.

Round 2
Reviewer 1 Report
The authors have partially taken into account the comments of review (for example, 1, 5, 7, 8, 10, 11, 15, etc.). They are given the opportunity to correct the entire publication in accordance with the comments made in the first review.
Point 10 about Fig.1. The author’s response: “The black square present in the upper right corner of each photo is the digital photographs of each sample”. Signature to Fig 10: “SEM surface micrographs of black MAO coating samples 1#-9#. The insertions: the digital photographs”.
According to Table 2, sample 6 is characterized by average values of porosity and thickness, and not excellent.
Surface porosity of the MAO coatings was measured by Image J software (line 163). It is impossible to draw an unambiguous conclusion about the difference in porosity according to Fig.5. The porosity values of the samples should be given.
